**∂ | Open Peer Review** | Epidemiology | Research Article

# Epidemiology of respiratory syncytial virus in young, hospitalized children in Jordan: a prospective viral surveillance study

Justin Z. Amarin,[1,2] Haya Hayek,[1] Olla Hamdan,[1] Yasmeen Z. Qwaider,[1] Tala Khraise,[1] Ahmad Khader,[1] Qusai Odeh,[1] Rami Salim,[1] Hadeel Shalabi,[1] Ahmad Alhajajra,[3] Yousef Khader,[4] Basim Al-Zoubi,[3] Najwa Khuri-Bulos,[5] Andrew J. Spieker,[6] Leigh M. Howard,[1] James D. Chappell,[1] Natasha B. Halasa,[1,5] for the Jordan Viral Surveillance Studies Group

**ABSTRACT** Respiratory syncytial virus (RSV) is a leading cause of hospitalization in young children. Understanding RSV burden and seasonality is crucial for implementing effective preventive strategies, especially in the wake of disruptions related to the coronavirus disease 2019 (COVID-19) pandemic. We aimed to determine RSV burden and seasonality among young children hospitalized in Jordan. We conducted a prospective viral surveillance study at Al-Bashir Hospital (1 November 2023 to 4 April 2024). Children <5 years old hospitalized with fever or respiratory symptoms were eligible. Nasal (and optional throat) swabs were collected and tested for common respiratory viruses using real-time polymerase chain reaction. We compared characteristics and outcomes of hospitalizations by RSV detection status and assessed RSV seasonality. Of 2,610 children, RSV was detected in 713 (27.3%), making it the second most common virus overall and the most common in children <2 years old ($n$ = 680 [30.0%]). Children with RSV were more likely than those without RSV to receive low-flow oxygen (74.9% vs 23.2%; $P < 0.001$) and high-flow nasal cannulation (3.2% vs 1.2%; $P < 0.001$) and were more likely to be admitted to the intensive care unit (13.2% vs 8.2%; $P < 0.001$). At least one other respiratory virus was co-detected with RSV in 244 children (34.2%). During the 2023–2024 season, RSV circulation exhibited a clear winter seasonality, consistent with historical patterns. In conclusion, the burden of RSV in children in Jordan remains substantial following the COVID-19 pandemic. The return to historical winter seasonality has important implications for the timing of preventive interventions. Continued surveillance is crucial for monitoring RSV epidemiology in this region.

**IMPORTANCE** This study confirms the persistent and significant burden of respiratory syncytial virus (RSV) among young, hospitalized children in Jordan. Crucially, our data reveal the normalization of RSV circulation patterns in 2024 following disruptions related to the coronavirus disease 2019 pandemic. This finding has important implications for optimizing the timing of preventive interventions, such as monoclonal antibodies and maternal vaccination, particularly in a resource-limited setting where they are costly and limited in availability. By providing these contemporary surveillance data from the Eastern Mediterranean—where sentinel surveillance platforms are lacking—this work has the potential to inform public health strategies directly and emphasizes the critical need for sustained monitoring to guide effective RSV prevention and control efforts.

**KEYWORDS** child, preschool, hospitalization, infant, Jordan, respiratory syncytial virus infections

**Peer Reviewers** Mustafa Kürşat Şahin, Ondokuz Mayıs University, Samsun, Atakum, Turkey; Pascal Lavoie, University of British Columbia, Vancouver, Canada

Address correspondence to Justin Z. Amarin, justin.amarin@vumc.org.

Justin Z. Amarin and Haya Hayek contributed equally to this article. The author order was determined alphabetically.

J.D.C. reports a current investigator-initiated grant from Merck. N.B.H. received grant support from Sanofi and Quidel, reports a current investigator-initiated grant from Merck, and serves on an advisory board for CSL Seqirus. The other authors did not report any conflict of interest.

See the funding table on p. 11.

Respiratory syncytial virus (RSV) is the most frequent and one of the most virulent respiratory pathogens during early childhood. Within the first 24 months of life, most children will have been infected with RSV, half of them twice (1). In 2019, RSV accounted for an estimated 33 million episodes of acute lower respiratory tract infection, 95% of which occurred in low- and middle-income countries (2). The virus drives a substantial proportion of pediatric hospitalizations worldwide; for example, in the United States, acute bronchiolitis due to RSV is the leading cause of infant hospitalizations (3). While most infants who develop severe RSV disease are previously healthy, risk factors for severe disease include prematurity, young infancy, and underlying cardiopulmonary, neurologic, and immunocompromising conditions (4, 5). Therefore, preventive measures are necessary to safeguard young children—especially infants—from RSV and its complications.

In addition to everyday prevention (e.g., handwashing, breastfeeding), several pharmaceutical interventions are currently available to protect young children from severe RSV. These options include monoclonal antibodies (e.g., palivizumab, nirsevimab) and maternal vaccination (e.g., bivalent RSV prefusion F subunit vaccine) (6–8). Palivizumab is a once-monthly monoclonal antibody recommended during the RSV season for infants at higher risk of severe disease but not healthy term infants or otherwise healthy preterm infants. Palivizumab prophylaxis is not considered high-value healthcare for any group of infants because its high cost is associated with minimal health benefits (6). Nirsevimab is a single-dose monoclonal antibody that is a much more potent inhibitor of RSV than palivizumab *in vitro*, has a substantially longer half-life, and is relatively less costly (6, 7). Due to its favorable properties, nirsevimab prophylaxis is used more broadly in high-income countries: in the United States, for example, nirsevimab prophylaxis is recommended for infants <8 months old born during or entering their first RSV season and for children 8–19 months old at higher risk of severe disease entering their second season (7). Another approach, seasonal maternal vaccination, may also be used broadly; in the United States, it is recommended for pregnant persons during 32 through 36 weeks of gestation to prevent severe RSV disease in infants <6 months old (8). The effectiveness of these pharmaceutical interventions relies heavily on properly aligning the timing of administration with the seasonality of RSV circulation.

Despite their demonstrated efficacy, RSV monoclonal antibodies and maternal vaccines are costly and limited in availability, particularly in low- and middle-income countries, where they are needed most. As of April 2025, only palivizumab is available in Jordan. In a modeling study, Li et al. showed that seasonal dosing approaches in these supply-constrained settings may be more cost-effective and feasible (9). Therefore, in anticipation of broader access to these interventions and the approval and release of additional products, robust data on RSV epidemiology across diverse geographic settings are crucial, especially because the seasonality of RSV circulation varies by geographic and meteorologic factors (10). Building on our group's previous viral surveillance (2007, 2010–2013, and 2020) at the largest public hospital in Amman, Jordan—Al-Bashir Hospital—we launched a contemporary viral surveillance study in January 2023 for a duration of 16 months to determine the burden of RSV in young, hospitalized children, with the hypothesis that RSV remains the most frequent and virulent respiratory virus (11–13). In addition, we investigated potential disruptions to the seasonality of RSV following the coronavirus disease 2019 (COVID-19) pandemic compared with historical patterns.

## MATERIALS AND METHODS

### Study design and population

Between 11 January 2023 and 30 April 2024, we conducted a prospective viral surveillance study of children <5 years old hospitalized in Al-Bashir Hospital, a large public hospital that provides care for at least 50%–60% of children in Amman, the capital

of Jordan and its most populous city (12). According to 2023 estimates, Amman was home to 42.0% of Jordan's 11.5 million residents—more than double that of any of Jordan's governorates (12 in total) (14). Eligible children were those <5 years old at hospital admission who developed a fever or at least one respiratory symptom within the preceding 14 days and were screened within 72 h of admission. Respiratory symptoms qualifying for eligibility included cough, earache, runny nose or nasal congestion, sore throat, posttussive vomiting, wheezing, rapid or shallow breathing, chest retractions or abdominal breathing, stridor, apnea (including brief resolved unexplained events), and myalgia. We excluded newborns never discharged home after birth, children previously enrolled within the 14 days preceding the day of hospital admission, and those with a known nonrespiratory cause for hospitalization. Research staff screened and enrolled eligible children 6 days per week (Saturday through Thursday), excluding public holidays and observances.

## Study procedures

After obtaining written informed consent from a parent or legal guardian, bilingual research staff administered an English-language standardized questionnaire to the parent or legal guardian. The staff was trained to convey questionnaire content verbally in Arabic, ensuring an accurate representation of the original English items. Following discharge, research staff abstracted additional data from electronic health records using a standardized case report form that included sections on health status at admission (e.g., results of chest radiography), review of underlying medical conditions, clinical course (e.g., intensive care unit admission, respiratory support, antibiotic use, and in-hospital mortality), and bacterial cultures and viral testing.

In addition, research staff collected an anterior nasal flocked swab from the child with or without a throat swab. The nasal swab, combined with the throat swab (if collected), was placed into a single vial containing PrimeStore Molecular Transport Medium or UTM Universal Transport Medium, stored at 4°C for a maximum of 28 days (PrimeStore) or −80°C (UTM), and shipped to Vanderbilt University Medical Center on frozen cold packs (PrimeStore) or dry ice (UTM). For a single participant, a throat swab was collected without an accompanying nasal swab. This specimen was processed and analyzed in the same manner as other specimens.

Laboratory staff at Vanderbilt University Medical Center tested specimens by singleplex reverse transcription real-time polymerase chain reaction assays for the following respiratory viruses: RSV; influenza virus types A, B, and C; parainfluenza virus (PIV) types 1–4; human metapneumovirus (HMPV); human rhinovirus (HRV); enterovirus D68 (EV-D68); adenovirus (AdV); common cold coronavirus (ccCoV) species 229E, HKU, NL63, and OC43; and severe acute respiratory syndrome coronavirus 2 (SARS-CoV-2). Notably, the polymerase chain reaction assay for HRV is cross-reactive with the genetically similar enteroviruses. Therefore, while we use the shorthand "HRV" to denote positive results, we acknowledge that some detections may represent enteroviruses.

## Statistical methods

We summarized categorical variables using absolute and relative frequencies and continuous variables using median and interquartile range. We used logistic regression for binary outcomes and linear regression for continuous outcomes to compare characteristics of children by RSV detection status, with RSV detection status as the predictor. Cluster-robust standard errors were used to account for children who enrolled more than once.

To assess the associations between RSV detection status and in-hospital outcomes, we used generalized estimating equations with a working independence structure, adjusting for well-established risk factors for severe RSV illness: namely, age at admission (fitted using restricted cubic splines with three knots), presence of at least one underlying medical condition, and prematurity status.

Statistical significance was determined to be achieved based on a nominal threshold of $\alpha = 0.05$ (two-sided, where applicable). All analyses were performed using R (version 4.4.1).

## RESULTS

### Study participants

Between 8 January 2023 and 30 April 2024, 9,252 children <18 years old were admitted to Al-Bashir Hospital, averaging 19.3 daily admissions (median, 19; range, 4–43) over 479 days. Of all children, 6,918 were <5 years old, representing 74.8% of the pediatric caseload. Our research staff began screening on 11 January 2023 and conducted screening activities on 393 (82.6%) of the 476 possible days. Research staff did not conduct screening on scheduled days off: Fridays (67 days), national holidays (15 days), and a general strike (1 day). The flow of study participants is shown in Fig. 1. Briefly, 3,128 children were provisionally eligible, 2,622 (83.8%) were enrolled, and 2,615 (99.7%) comprised the study population. To construct our analytic sample, we excluded children with no respiratory specimen available for molecular diagnostics, yielding an analytic sample size of 2,610. The sample included 2,408 children, with 2,238 enrolled once ($n = 2,238$ observations; 85.7%), 142 enrolled twice ($n = 284$ observations; 10.9%), 24 enrolled three times ($n = 72$ observations; 2.8%), and 4 enrolled four times ($n = 16$ observations; 0.6%). Throughout the article, we use the term "children" as shorthand to refer to discrete hospitalizations, while accounting for clustered observations in all statistical comparisons.

### Respiratory virus detections

Of the 2,610 children, 2,121 (81.3%) were positive for at least one respiratory virus (Table S1). The most common virus was HRV ($n = 733$ [28.1%]), followed by RSV ($n = 713$ [27.3%]) and AdV ($n = 325$ [12.5%]). In children <2 years old, RSV was the most common virus ($n = 680$ [30.0%]). Notably, 608 out of 713 children with RSV (85.3%) were <1 year old. The proportions of RSV detection among subgroups defined by age (in months) are shown in Fig. S1. Overall, RSV was co-detected with at least one other respiratory virus in 244 children (34.2%). The most common co-detected virus was HRV ($n = 112$ [15.7%]), followed by ccCoV ($n = 51$ [7.2%]), SARS-CoV-2 ($n = 34$ [4.8%]), AdV ($n = 32$ [4.5%]), influenza virus ($n = 27$ [3.8%]), PIV ($n = 23$ [3.2%]), and HMPV ($n = 14$ [2.0%]).

### Seasonality of RSV

During the study period, RSV was detected in a seasonal pattern, peaking in the winter of 2024 (Fig. 2a). RSV detections during the midwinter and late winter of 2023 appeared to have been trailing off, suggesting that the peak had occurred in late 2022. Indeed, a plot of the monthly circulation of RSV per annum in the same setting (using historical surveillance data for reference; Fig. 2b) showed that RSV circulation in the first quarter of 2023 was atypical, whereas the circulation of RSV during 2024 was consistent with historical patterns (11–13).

### Baseline characteristics

We compared baseline characteristics of children with or without RSV (Table 1). Briefly, children with RSV tended to be younger (median, 2.9 months vs 5.7 months; $P < 0.001$) and were less likely to be male (52.2% vs 56.9%; $P = 0.028$), less likely to have at least one underlying medical condition (21.2% vs 30.7%; $P < 0.001$), more likely to be breastfed at the time of enrollment (61.0% vs 47.5%; $P < 0.001$), less likely to attend daycare or preschool (2.4% vs 4.0%; $P = 0.047$), less likely to be exposed to any tobacco-related smoke (80.8% vs 84.3%; $P = 0.030$), and more likely to receive antibiotics prior to

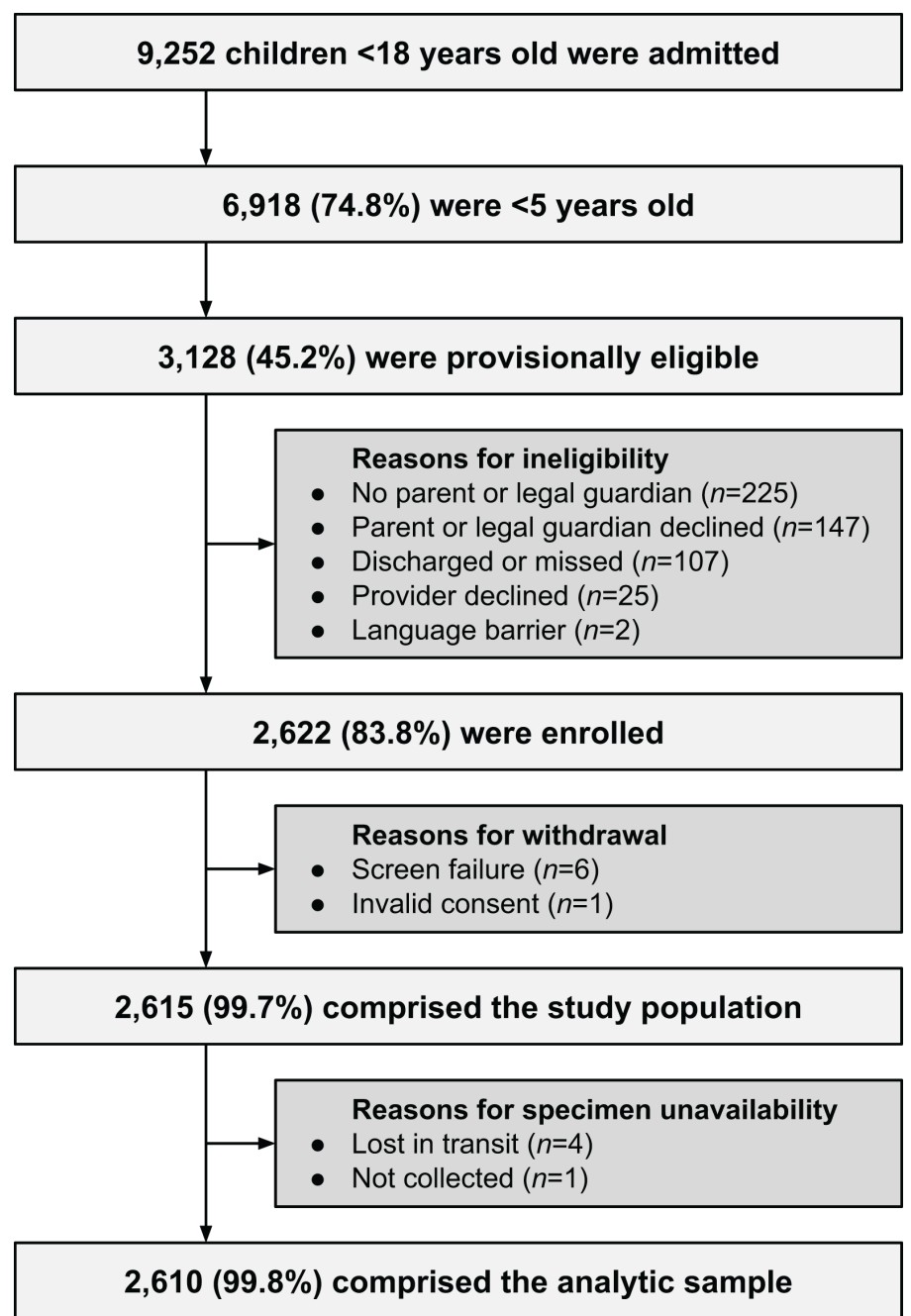

**FIG 1** Flow diagram of study participants.

hospitalization (38.0% vs 33.6%; *P* = 0.037). Overall, 5 out of 2,594 children (0.2%) received palivizumab prior to hospitalization.

## Symptoms and diagnostics

The burden of symptoms at or prior to hospital admission tended to be higher among children with RSV, with some exceptions. Of the 21 signs and symptoms we evaluated, the frequencies of 10 were significantly higher among children with RSV, while the frequencies of 5 were significantly lower (Fig. 3). Notably, 13 out of 2,610 children (0.5%) were clinically tested for RSV prior to or within the first 72 h of admission (Table 1).

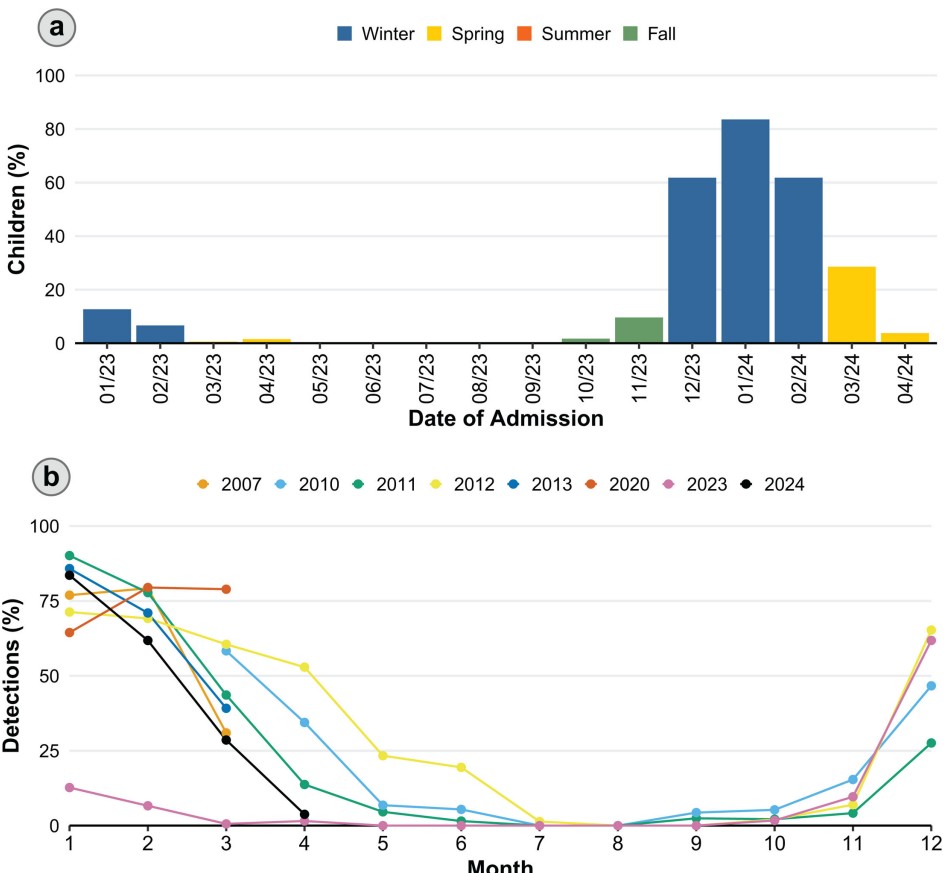

**FIG 2** Circulation of respiratory syncytial virus (RSV) in Amman, Jordan. (a) Seasonality of RSV detected in children <5 years old hospitalized at Al-Bashir Hospital (Amman, Jordan) with fever or respiratory symptoms between 8 January 2023 and 30 April 2024. (b) Monthly detections of RSV per annum in children hospitalized at Al-Bashir Hospital (Amman, Jordan) with fever or respiratory symptoms (2007, 2010–2013, 2020, and 2023–2024). Children included in the 2007 and 2023–2024 studies were <5 years old, while those in the 2010–2013 and 2020 studies were <2 years old.

## In-hospital outcomes

Children with RSV were more likely than those without RSV to receive antibiotics during hospitalization (97.6% vs 94.3%; $P < 0.001$), low-flow oxygen (74.9% vs 23.2%; $P < 0.001$), and high-flow nasal cannulation (3.2% vs 1.2%; $P < 0.001$). They were also more likely to be admitted to the intensive care unit (13.2% vs 8.2%; $P < 0.001$; Table 1). We present select in-hospital outcomes among subgroups of children defined more granularly by detection status in Table 2, and we present the adjusted associations between RSV detection status and in-hospital outcomes in Table 3. The results of these adjusted analyses were consistent with those of the unadjusted analyses. The odds of antibiotic use during hospitalization were 2.03 higher (95% confidence interval [95% CI], 1.20–3.41) among children with RSV than those without. The odds of low-flow oxygen were also significantly higher (adjusted odds ratio [aOR], 10.0; 95% CI, 8.2–12.4) among children with RSV than those without, as were the odds of high-flow nasal cannula (aOR, 2.60; 95% CI, 1.44–4.69) and intensive care unit admission (aOR, 1.65; 95% CI, 1.25–2.17).

## DISCUSSION

In this prospective viral surveillance study conducted between 11 January 2023 and 30 April 2024, at Jordan's largest public hospital, RSV was the second most common viral pathogen among children <5 years old hospitalized with fever or respiratory symptoms and the most common among those <2 years old, especially during infancy. Children

**TABLE 1** Baseline characteristics, diagnostics, and in-hospital outcomes of children <5 years old hospitalized at Al-Bashir Hospital (Amman, Jordan) with fever or respiratory symptoms between 8 January 2023 and 30 April 2024, stratified by RSV detection status (by research testing)[a]

| Parameter | Overall, N = 2,610 | RSV detection status | | P-value[b] |
|---|---|---|---|---|
| | | Not detected, n = 1,897 | Detected, n = 713 | |
| Baseline characteristics | | | | |
| Age at admission (months)—median (IQR) | 4.5 (1.7–13.0) | 5.7 (1.9–16.5) | 2.9 (1.4–6.7) | **<0.001** |
| Male—n (%) | 1,452 (55.6) | 1,080 (56.9) | 372 (52.2) | **0.028** |
| At least one underlying medical condition—n (%) | 733 (28.1) | 582 (30.7) | 151 (21.2) | **<0.001** |
| Born prematurely[c]—n (%) | 434 (16.6) | 313 (16.5) | 121 (17.0) | 0.77 |
| Currently breastfed—n (%) | 1,337 (51.2) | 902 (47.5) | 435 (61.0) | **<0.001** |
| Daycare or preschool attendance—n (%) | 93/2,608 (3.6) | 76/1,896 (4.0) | 17/712 (2.4) | **0.047** |
| Any tobacco-related smoke exposure—n (%) | 2,176 (83.4) | 1,600 (84.3) | 576 (80.8) | **0.030** |
| Antibiotic use prior to hospitalization—n (%) | 902/2,592 (34.8) | 632/1,881 (33.6) | 270/711 (38.0) | **0.037** |
| Palivizumab use prior to hospitalization—n (%) | 5/2,594 (0.2) | 5/1,882 (0.3) | 0/712 (0.0) | NA[d] |
| Diagnostics prior to or within 72 h of admission | | | | |
| Clinical testing for RSV—n (%) | 13 (0.5) | 3 (0.2) | 10 (1.4) | NA[d] |
| Detected—n (%) | 10/13 (76.9) | 0/3 (0.0) | 10/10 (100.0) | |
| Clinical testing for influenza virus—n (%) | 8 (0.3) | 2 (0.1) | 6 (0.8) | NA[d] |
| Detected—n (%) | 0/8 (0.0) | 0/2 (0.0) | 0/6 (0.0) | |
| Clinical testing for SARS-CoV-2—n (%) | 13 (0.5) | 9 (0.5) | 4 (0.6) | NA[d] |
| Detected—n (%) | 1/13 (7.7) | 1/9 (11.1) | 0/4 (0.0) | |
| In-hospital outcomes | | | | |
| Antibiotic use during hospitalization—n (%) | 2,484 (95.2) | 1,788 (94.3) | 696 (97.6) | **<0.001** |
| Days in hospital—median (IQR) | 4.0 (3.0–6.0) | 4.0 (3.0–6.0) | 5.0 (3.0–7.0) | 0.45 |
| Low-flow oxygen—n (%) | 974 (37.3) | 440 (23.2) | 534 (74.9) | **<0.001** |
| Days on low-flow oxygen—median (IQR) | 3.0 (1.3–5.0) | 2.0 (1.0–5.0) | 4.0 (2.0–6.0) | 0.051 |
| High-flow nasal cannula—n (%) | 46 (1.8) | 23 (1.2) | 23 (3.2) | **<0.001** |
| Days on high-flow nasal cannula—median (IQR) | 4.0 (3.0–6.0) | 4.0 (2.5–7.0) | 3.0 (3.0–5.0) | 0.095 |
| CPAP/BiPAP—n (%) | 27 (1.0) | 19 (1.0) | 8 (1.1) | 0.79 |
| Days on CPAP/BiPAP—median (IQR) | 3.0 (1.0–3.0) | 2.0 (1.0–3.5) | 3.0 (1.8–3.0) | 0.90 |
| Intensive care unit admission—n (%) | 249 (9.5) | 155 (8.2) | 94 (13.2) | **<0.001** |
| Invasive mechanical ventilation—n (%) | 18 (0.7) | 16 (0.8) | 2 (0.3) | NA[d] |
| Days mechanically ventilated—median (IQR) | 6.0 (3.3–8.8) | 6.0 (2.8–9.3) | 5.5 (5.3–5.8) | 0.57 |
| Death—n (%) | 12 (0.5) | 11 (0.6) | 1 (0.1) | NA[d] |

[a]RSV, respiratory syncytial virus; IQR, interquartile range; NA, not applicable; SARS-CoV-2, severe acute respiratory syndrome coronavirus 2; CPAP, continuous positive airway pressure; BiPAP, bilevel positive airway pressure.
[b]P-values are derived from logistic (binary outcomes) or linear (continuous outcomes) regression with cluster-robust standard errors. Values in bold denote statistical significance based on a nominal threshold of α = 0.05.
[c]Prematurity status was determined based on the conventional cutoff of <37 weeks of gestational age.
[d]We opted not to compute inferential statistics because of data sparsity (i.e., any expected cell count < 5).

with RSV, who represented nearly one-third of the study population, generally had a more severe course of illness than children without RSV, as indicated by higher proportions of respiratory support and intensive care utilization. Co-detection of RSV with other respiratory viruses—particularly HRV—was common, occurring in roughly one-third of children with RSV. Importantly, our surveillance uniquely captured the apparent normalization of RSV circulation patterns in 2024 following disruptions related to the COVID-19 pandemic, with winter peaks aligning with historical seasonal trends observed in this setting in the past.

The substantial burden of RSV we observed among hospitalized, young children in Amman, Jordan, concords with global estimates highlighting RSV as the leading cause of acute lower respiratory infections in this age group (2). Our finding that children with RSV required respiratory support and intensive care at higher proportions than their counterparts without RSV echoes findings from other settings, including our prior viral surveillance studies in Amman, Jordan (11, 12, 15–19). In a viral surveillance study conducted in Nashville, TN, Haddadin et al. evaluated the severity of RSV in children <5 years old with fever or respiratory symptoms and found that 61.5% and 16.9% required

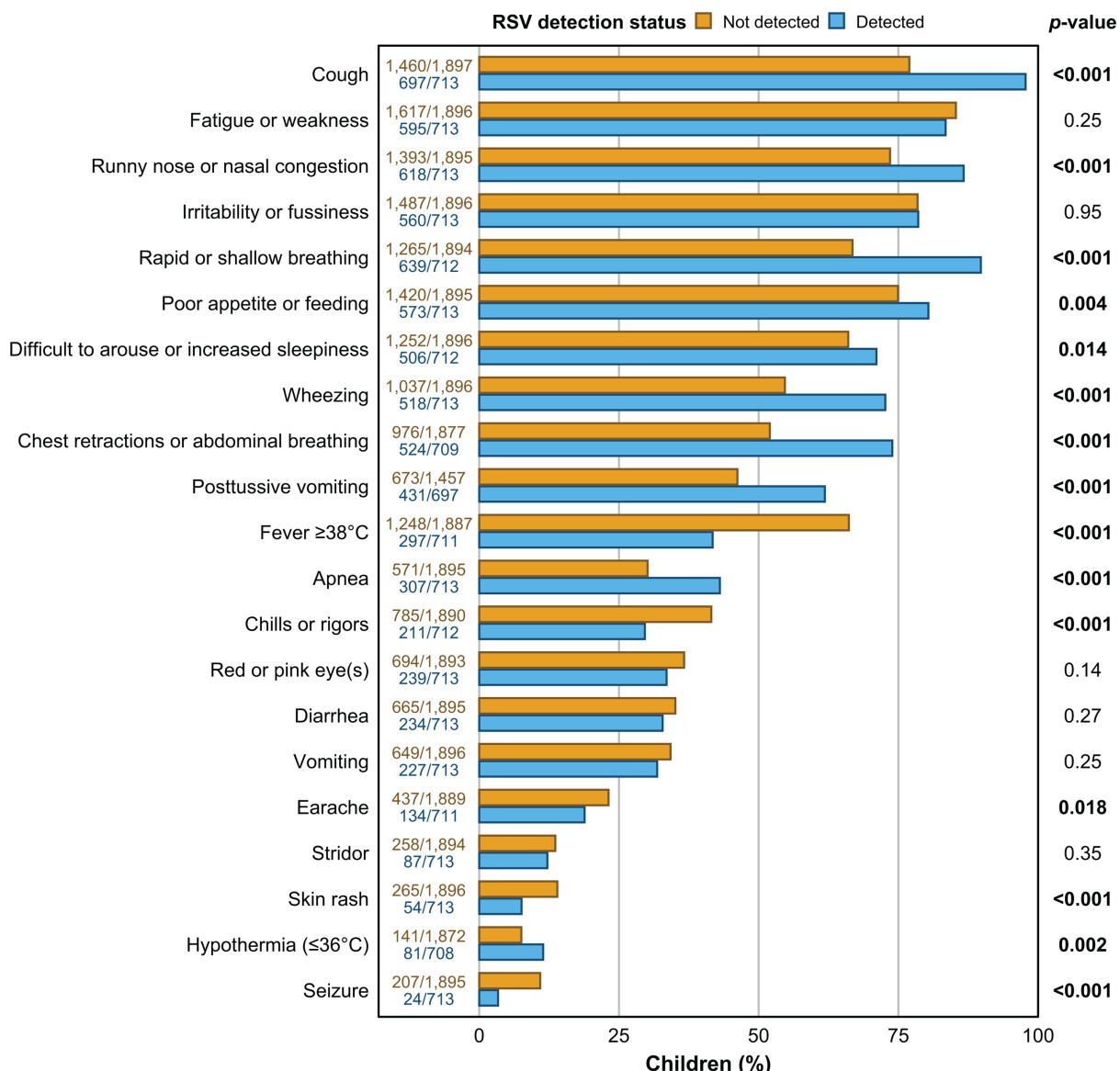

**FIG 3** Symptoms at or prior to hospital admission among children <5 years old hospitalized at Al-Bashir Hospital (Amman, Jordan) with fever or respiratory symptoms between 8 January 2023 and 30 April 2024, stratified by respiratory syncytial virus (RSV) detection status.

supplemental oxygen and intensive care, respectively, consistent with our results (74.9% and 13.2%, respectively) (15). In both cohorts, the clinical presentation of RSV was characterized by higher proportions of lower respiratory symptoms—particularly rapid or shallow breathing, wheezing, and chest retractions—consistent with RSV's predilection for the lower airways (17). Despite this characteristic presentation and severity profile, we found that clinical testing for RSV was rarely performed (only 0.5% of children were tested). This lack of routine testing likely contributes to diagnostic uncertainty and may explain, at least partially, the near-ubiquitous use of in-hospital antibiotics. In complementary analyses of this and prior surveillance cohorts at Al-Bashir Hospital, we showed that over 90% of children received antibiotics despite viral etiologies predominating (20). In low-resource settings, the downstream impact of such diagnostic gaps is amplified, fueling widespread antibiotic use and contributing to antimicrobial resistance. Enhancing rapid diagnostic capacity is, therefore, essential not only for optimizing individual patient management and promoting antimicrobial stewardship but also for robustly monitoring the ongoing effectiveness of early preventive interventions.

**TABLE 2** In-hospital outcomes among subgroups of children (defined by respiratory viral detection) hospitalized at Al-Bashir Hospital (Amman, Jordan) with fever or respiratory symptoms between 8 January 2023 and 30 April 2024[a]

| Virus[b] | Low-flow oxygen | High-flow nasal cannula | CPAP/BiPAP | ICU admission | IMV | Death |
|---|---|---|---|---|---|---|
| HRV | 253 (34.5) | 11 (1.5) | 12 (1.6) | 72 (9.8) | 8 (1.1) | 3 (0.4) |
| RSV | 534 (74.9) | 23 (3.2) | 8 (1.1) | 94 (13.2) | 2 (0.3) | 1 (0.1) |
| AdV | 87 (26.8) | 3 (0.9) | 4 (1.2) | 28 (8.6) | 3 (0.9) | 1 (0.3) |
| HMPV | 88 (34.5) | 3 (1.2) | 5 (2.0) | 25 (9.8) | 3 (1.2) | 1 (0.4) |
| PIV | 59 (27.3) | 1 (0.5) | 4 (1.9) | 16 (7.4) | 3 (1.4) | 1 (0.5) |
| ccCoV | 67 (32.4) | 2 (1.0) | 2 (1.0) | 15 (7.2) | 0 (0.0) | 0 (0.0) |
| SARS-CoV-2 | 48 (25.0) | 2 (1.0) | 2 (1.0) | 10 (5.2) | 2 (1.0) | 2 (1.0) |
| Flu | 36 (20.3) | 2 (1.1) | 0 (0.0) | 4 (2.3) | 0 (0.0) | 0 (0.0) |

[a]CPAP, continuous positive airway pressure; BiPAP, bilevel positive airway pressure; ICU, intensive care unit; IMV, invasive mechanical ventilation; HRV, human rhinovirus; RSV, respiratory syncytial virus; AdV, adenovirus; HMPV, human metapneumovirus; PIV, parainfluenza virus; ccCoV, common cold coronavirus; SARS-CoV-2, severe acute respiratory syndrome coronavirus 2; Flu, influenza virus.
[b]Children with viral co-detection are included; subgroups are not mutually exclusive.

Moreover, the young age distribution of children in whom RSV was detected in our study, with most in infancy, is congruent with a well-established vulnerability window and supports current recommendations for early preventive interventions, including maternal vaccination during pregnancy and nirsevimab administration during early life (6–8).

Our observation that RSV circulation in Amman has returned to its historical winter seasonality following pandemic-related disruptions mirrors emerging global patterns and has important implications for the timing of preventive interventions. A recent multinational analysis demonstrated that while initial RSV epidemics following the COVID-19 pandemic exhibited unusual temporality, subsequent waves progressively resembled typical RSV seasonality across geographic settings, albeit with little representation from the Eastern Mediterranean region, where sentinel surveillance platforms are lacking (21). In Jordan—and other resource-limited settings—understanding these consistent seasonal patterns is crucial for optimizing the timing of preventive measures (9). The close monitoring we performed of circulation onset, peak, and offset will be particularly valuable for informing the delivery of preventive interventions as global RSV dynamics continue to stabilize. Beyond seasonality, we also observed a high proportion of viral co-detection, mirroring findings from other recent studies. For instance, our

**TABLE 3** Estimates from generalized estimating equations (GEE) models of in-hospital outcomes comparing children <5 years old hospitalized at Al-Bashir Hospital (Amman, Jordan) with fever or respiratory symptoms, with or without RSV, between 8 January 2023 and 30 April 2024[a]

| Outcome | Point estimate (95% CI) | P-value[b] |
|---|---|---|
| Antibiotic use during hospitalization—aOR | 2.03 (1.20–3.41) | **0.008** |
| Days in hospital—aMR | −0.32 (−0.79–0.14) | 0.17 |
| Low-flow oxygen—aOR | 10.0 (8.2–12.4) | **<0.001** |
| Days on low-flow oxygen—aMR | 0.38 (−0.06–0.82) | 0.092 |
| High-flow nasal cannula—aOR | 2.60 (1.44–4.69) | **0.001** |
| Days on high-flow nasal cannula—aMR | −1.24 (−2.98–0.49) | 0.17 |
| CPAP/BiPAP—aOR | 0.98 (0.43–2.21) | 0.95 |
| Days on CPAP/BiPAP—aMR | −0.09 (−2.52–2.34) | 0.94 |
| Intensive care unit admission—aOR | 1.65 (1.25–2.17) | **<0.001** |
| Days mechanically ventilated—aMR | 0.05 (−5.13–5.24) | 0.98 |

[a]RSV, respiratory syncytial virus; 95% CI, 95% confidence interval; aOR, adjusted odds ratio; aMR, adjusted mean difference; CPAP, continuous positive airway pressure; BiPAP, bilevel positive airway pressure.
[b]P-values are derived from logistic regression for binary outcomes and linear regression for continuous outcomes. All estimates are adjusted for age at admission (fitted using restricted cubic splines with three knots), presence of at least one underlying medical condition (binary), and prematurity status (binary). In all comparisons, children without RSV are the reference group. Values in bold denote statistical significance based on a nominal threshold of $\alpha = 0.05$.

group previously reported that 39.0% of children <5 years old with RSV presenting to outpatient, emergency department, or inpatient settings in a single medical center in Nashville, TN (2018–2022), were positive for one other common respiratory virus (by clinical testing), consistent with our estimate of 34.2% (16). While the high proportion of viral co-detection raises important considerations about the interpretation of positive respiratory viral tests during peak RSV season, the clinical significance of RSV co-detection remains to be fully elucidated (22). Nevertheless, these cases highlight the complex viral ecology that characterizes pediatric respiratory infections during winter months and emphasize the need for ongoing research.

While global data highlight the evolving dynamics of RSV, studies conducted within Jordan by other research groups consistently reinforce the substantial burden and characteristic seasonality observed in our investigation. Notably, investigations of RSV in Jordan date back as early as 1993 (23), providing a long-term perspective on the epidemiology of the respiratory virus in the country. Earlier studies at hospitals in Amman, including Al-Bashir Hospital, prior to the COVID-19 pandemic, reported similar RSV detection proportions among hospitalized children with respiratory tract infections, as well as a comparable winter seasonality (24, 25). These findings are further grounded by similar results from studies performed in northern Jordan and another governorate in central Jordan, suggesting a high degree of consistency in RSV epidemiology across different regions of the country (23, 26, 27). Studies encompassing periods of COVID-19-related disruption in circulation—with some relying on clinical testing rather than systematic surveillance—provide additional context (28, 29), and a more recent multicenter surveillance study conducted between November 2022 and April 2023 using a geographically representative sample supports the general burden of RSV characterized herein (30). Importantly, many of these studies, particularly those utilizing clinical data, help to fill temporal gaps in our continual surveillance efforts (28–31). The consistency of findings across multiple studies, spanning various periods (from 1993 to 2023) and geographic settings (from the north to the south of Jordan), strengthens the validity of our findings and underscores the persistent public health challenge posed by RSV in Jordan. Collectively, these findings highlight the urgent need to implement effective RSV prevention measures to reduce the burden of RSV-related hospitalizations, especially given that medical care for children <6 years old in Jordan is government-funded. Our findings provide inputs for evaluating the cost-effectiveness of preventive measures such as nirsevimab and maternal vaccination. Although formal economic analyses are beyond the scope of this study, modeling work has shown that aligning administration with local seasonality can improve feasibility and value in resource-limited settings (9). Companion cost-effectiveness studies are warranted to fully evaluate the benefits of such measures.

Our study has several strengths. First, we provide robust viral surveillance data from the Eastern Mediterranean region, where sentinel surveillance platforms are lacking and contemporary RSV circulation dynamics are poorly understood. Second, our research group's history of conducting similarly designed studies at Al-Bashir Hospital since 2007 provides valuable context to assess current RSV circulation patterns in Jordan, fortifying our ability to identify true deviations from typical patterns. Our study also has several limitations. First, as a single-center study, our findings may not fully represent RSV circulation dynamics throughout Jordan, though Al-Bashir's large catchment area and Jordan's meteorologic homogeneity reasonably permit generalization to urban areas (as corroborated by other studies in diverse settings across Jordan). Nevertheless, RSV circulation in rural regions might differ. Second, while our historical data span multiple years, they are discontinuous, potentially missing important transitional periods. However, the consistency we observed in RSV seasonality across discrete study periods lends credibility to our characterization of "typical" dynamics in Jordan and supports our interpretation of post-pandemic normalization.

In conclusion, our findings highlight the importance of sustained viral surveillance in regions bearing a disproportionate public health burden from RSV, particularly as new

preventive interventions become available. While the normalization of RSV circulation following pandemic-related disruptions suggests a return to predictable seasonality, continued monitoring remains crucial for optimizing the timing of preventive strategies.

## ACKNOWLEDGMENTS

We gratefully acknowledge the laboratory support of Bryan Peterson, Claudia Guevara Pulido, Laura Short, and Shanice Cummings.

This study was supported, in part, by a research grant from the Investigator-Initiated Studies Program of Merck Sharp & Dohme LLC. The opinions expressed in this paper are those of the authors and do not necessarily represent those of Merck Sharp & Dohme LLC.

Conceptualization: J.Z.A., H.H., and N.B.H. Data curation: J.Z.A. Formal analysis: J.Z.A. and A.J.S. Funding acquisition: J.Z.A. and N.B.H. Investigation: T.K., A.K., Q.O., R.S., and H.S. Methodology: J.Z.A. and A.J.S. Project administration: J.Z.A., H.H., J.D.C., and N.B.H. Supervision: N.B.H. Validation: J.D.C. Visualization: J.Z.A. Writing—original draft: J.Z.A. and H.H. Writing—review & editing: J.Z.A., H.H., O.H., Y.Z.Q., T.K., A.K., Q.O., R.S., H.S., A.A., Y.K., B.A.-Z., N.K.-B., A.J.S., L.M.H., J.D.C., and N.B.H.

## AUTHOR AFFILIATIONS

[1]Division of Pediatric Infectious Diseases, Department of Pediatrics, Vanderbilt University Medical Center, Nashville, Tennessee, USA
[2]Epidemiology Doctoral Program, School of Medicine, Vanderbilt University, Nashville, Tennessee, USA
[3]Department of Pediatrics, Al-Bashir Hospital, Amman, Jordan
[4]Department of Public Health and Community Medicine, Faculty of Medicine, Jordan University of Science and Technology, Irbid, Jordan
[5]Department of Pediatrics, School of Medicine, The University of Jordan, Amman, Jordan
[6]Department of Biostatistics, Vanderbilt University Medical Center, Nashville, Tennessee, USA

## AUTHOR ORCIDs

Justin Z. Amarin  http://orcid.org/0000-0002-4484-1077
Haya Hayek  http://orcid.org/0000-0003-0087-4392
Olla Hamdan  http://orcid.org/0000-0002-3703-5536
Leigh M. Howard  http://orcid.org/0000-0003-4301-7235

## FUNDING

| Funder | Grant(s) | Author(s) |
|---|---|---|
| Merck & Co. | | Natasha B. Halasa |

## AUTHOR CONTRIBUTIONS

Justin Z. Amarin, Conceptualization, Data curation, Formal analysis, Funding acquisition, Methodology, Project administration, Visualization, Writing – original draft, Writing – review and editing | Haya Hayek, Conceptualization, Project administration, Writing – original draft, Writing – review and editing | Olla Hamdan, Writing – review and editing | Yasmeen Z. Qwaider, Writing – review and editing | Tala Khraise, Investigation, Writing – review and editing | Ahmad Khader, Investigation, Writing – review and editing | Qusai Odeh, Investigation, Writing – review and editing | Rami Salim, Investigation, Writing – review and editing | Hadeel Shalabi, Investigation, Writing – review and editing | Ahmad Alhajajra, Writing – review and editing | Yousef Khader, Writing – review and editing | Basim Al-Zoubi, Writing – review and editing | Najwa Khuri-Bulos, Writing – review and editing | Andrew J. Spieker, Formal analysis, Methodology, Writing – review and editing

| Leigh M. Howard, Writing – review and editing | James D. Chappell, Project administration, Validation, Writing – review and editing | Natasha B. Halasa, Conceptualization, Funding acquisition, Project administration, Supervision, Writing – review and editing

## ETHICS APPROVAL

The Vanderbilt University Institutional Review Board and the Jordanian Ministry of Health Institutional Review Board reviewed and approved the study protocol. Parents or legal guardians provided written informed consent.

## ADDITIONAL FILES

The following material is available online.

### Supplemental Material

**Supplemental material (Spectrum01727-25-S0001.docx).** Table S1; Fig. S1; Jordan Viral Surveillance Studies Group member list.

### Open Peer Review

**PEER REVIEW HISTORY (review-history.pdf).** An accounting of the reviewer comments and feedback.

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
