## [Reviewer comments · Microbiology Spectrum]

Microbiology Spectrum

Epidemiology of respiratory syncytial virus in young, hospitalized children in Jordan: A prospective viral surveillance study

Justin Amarin, Haya Hayek, Olla Hamdan, Yasmeen Qwaider, Tala Khraise, Ahmad Khader, Qusai Odeh, Rami Salim, Hadeel Shalabi, Ahmad Alhajajra, Yousef Khader, Basim Al-Zoubi, Najwa Khuri-Bulos, Andrew Spieker, Leigh Howard, James Chappell, and Natasha Halasa

Corresponding Author(s): Justin Amarin, Vanderbilt University

Review Timeline:

Submission Date:	June 4, 2025
Editorial Decision:	August 14, 2025
Revision Received:	September 4, 2025
Accepted:	September 22, 2025

Editor: Alex Dulovic

Reviewer(s): Disclosure of reviewer identity is with reference to reviewer comments included in decision letter(s). The following individuals involved in review of your submission have agreed to reveal their identity: Mustafa Kürşat Şahin (Reviewer #1); Pascal Lavoie (Reviewer #2)

Transaction Report:

DOI: <https://doi.org/10.1128/spectrum.01727-25>

Re: Spectrum01727-25 (Epidemiology of respiratory syncytial virus in young, hospitalized children in Jordan: A prospective viral surveillance study)

Dear Dr. Justin Z. Amarin:

Thank you for the privilege of reviewing your work. Below you will find my comments, instructions from the Spectrum editorial office, and the reviewer comments.

Based upon reviewers comments and my own review of the manuscript, my decision is MODIFICATIONS.

Revision Guidelines

Sincerely,
Alex Dulovic
Editor
Microbiology Spectrum

Reviewer #1 (Comments for the Author):

This manuscript presents a high-quality, meticulously executed, and policy-relevant viral surveillance study of respiratory syncytial virus (RSV) burden in Jordanian children under 5 years of age. The study's strengths lie in its prospective design, robust sample size, and integration of clinical, epidemiologic, and laboratory data. It fills an important evidence gap in the Eastern Mediterranean region, where systematic RSV surveillance is sparse. The findings have strong implications for the timing

and implementation of prophylactic measures, especially given the post-COVID-19 resurgence and normalization of RSV seasonality. A few suggestions for strengthening the manuscript:

Statistical Modeling: Consider presenting adjusted odds ratios for major in-hospital outcomes (e.g., ICU admission, oxygen use), accounting for age, prematurity, and comorbidities.

Diagnostic Stewardship: Expand briefly in the discussion on the underutilization of RSV-specific diagnostic testing (only 0.5% tested clinically), and how this may contribute to high antibiotic use. This point is crucial in low-resource settings.

Cost and Policy Considerations: A short comment on the potential cost-effectiveness of implementing RSV preventive measures (e.g., nirsevimab or maternal vaccination), even at the modeling level, would add public health value.

Figure Suggestions: Improve the clarity of Figure 3 (symptom burden) by better differentiating statistically significant differences, and consider increasing font size for accessibility.

Reviewer #2 (Comments for the Author):

This prospective surveillance study described the viral epidemiology of children under 5 years of age admitted to the main hospital in Amman, the capital of Jordan. The data in this understudied region of the middle east incrementally add to pre-pandemic epidemiological reports. The study is generally well reported and written. The findings are expected. The study description would be improved by clarifying how the symptoms list were collected (consider adding the standardized questionnaire as a supplemental file) and over what period. For instance, if fever was an inclusion criteria how come only a proportion of children report fever by RSV detection status in Figure 3?

Also, the authors may want to consider omitting p values for the baseline variables in Table 1, which are superfluous and not necessarily reflecting meaningful clinical differences in some cases (e.g. exposure to smoking 84 vs 81%) within the sample size.

This manuscript presents a high-quality, meticulously executed, and policy-relevant viral surveillance study of respiratory syncytial virus (RSV) burden in Jordanian children under 5 years of age. The study's strengths lie in its prospective design, robust sample size, and integration of clinical, epidemiologic, and laboratory data. It fills an important evidence gap in the Eastern Mediterranean region, where systematic RSV surveillance is sparse. The findings have strong implications for the timing and implementation of prophylactic measures, especially given the post-COVID-19 resurgence and normalization of RSV seasonality. A few suggestions for strengthening the manuscript:

Statistical Modeling: Consider presenting adjusted odds ratios for major in-hospital outcomes (e.g., ICU admission, oxygen use), accounting for age, prematurity, and comorbidities.

Response 1: Thank you for the suggestion. We agree that assessing the association between RSV detection status and in-hospital outcomes while adjusting for these well-established risk factors for severe RSV illness would be valuable. We performed these analyses and now report the results, which were consistent with the unadjusted results, in **Table 3**.

We have added the following description to the “*Statistical Methods*” subsection of the “*Methods*” section:

“To assess the associations between RSV detection status and in-hospital outcomes, we used generalized estimating equations with a working independence structure, adjusting for well-established risk factors for severe RSV illness: namely, age at admission (fitted using restricted cubic splines with three knots), presence of at least one underlying medical condition, and prematurity status.”

We have also added the following text to the “*In-Hospital Outcomes*” subsection of the “*Results*” section:

“...and we present the adjusted associations between RSV detection status and in-hospital outcomes in Table 3. The results of these adjusted analyses were consistent with those of the unadjusted analyses. The odds of antibiotic use during hospitalization were 2.03 higher (95% confidence interval [95% CI], 1.20–3.41) among children with RSV than those without. The odds of low-flow oxygen were also significantly higher (adjusted odds ratio [aOR], 10.0; 95% CI, 8.2–12.4) among children with RSV than those without, as were the odds of high-flow nasal cannula (aOR, 2.60; 95% CI, 1.44–4.69) and intensive care unit admission (aOR, 1.65; 95% CI, 1.25–2.17).”

Diagnostic Stewardship: Expand briefly in the discussion on the underutilization of RSV-specific diagnostic testing (only 0.5% tested clinically), and how this may contribute to high antibiotic use. This point is crucial in low-resource settings.

Response 2: We have expanded the relevant paragraph and added a new reference. The revised text now reads as follows (with additions underlined):

“Despite this characteristic presentation and severity profile, we found that clinical testing for RSV was rarely performed (only 0.5% of children were tested). This lack of routine testing likely contributes to diagnostic uncertainty and may explain, at least partially, the near-ubiquitous use of in-hospital antibiotics. In complementary analyses of this and prior surveillance cohorts at Al-Bashir Hospital, we showed that over 90% of children received antibiotics despite viral etiologies predominating [20]. In low-resource settings, the downstream impact of such diagnostic gaps is amplified, fueling widespread antibiotic use and contributing to antimicrobial resistance. Enhancing rapid diagnostic capacity is therefore essential not only for optimizing individual patient management and promoting antimicrobial stewardship but also for robustly monitoring the ongoing effectiveness of early preventive interventions.”

Cost and Policy Considerations: A short comment on the potential cost-effectiveness of implementing RSV preventive measures (e.g., nirsevimab or maternal vaccination), even at the modeling level, would add public health value.

Response 3: Thank you for the important suggestion. We have added the following text to the “Discussion” section:

“Our findings provide inputs for evaluating the cost-effectiveness of preventive measures such as nirsevimab and maternal vaccination. Although formal economic analyses are beyond the scope of this study, modeling work has shown that aligning administration with local seasonality can improve feasibility and value in resource-limited settings [9].”

Figure Suggestions: Improve the clarity of Figure 3 (symptom burden) by better differentiating statistically significant differences, and consider increasing font size for accessibility.

Response 4: Thank you for the feedback. We designed **Figure 3** with accessibility in mind. We maximized font sizes within the constraints of the maximum dimensions recommended by *Microbiology Spectrum*, and we denoted statistical significance by bolding the relevant values. We also used the color palette recommended by Wong (*Nat Methods*. 2011; 8(6): 441) to make the figure accessible to people with color vision deficiency. We are happy to make additional changes to this figure at the recommendations of the editorial board.

Reviewer 2

This prospective surveillance study described the viral epidemiology of children under 5 years of age admitted to the main hospital in Amman, the capital of Jordan. The data in this understudied region of the middle east incrementally add to pre-pandemic epidemiological reports. The study is generally well reported and written. The findings are expected.

The study description would be improved by clarifying how the symptoms list were collected (consider adding the standardized questionnaire as a supplemental file) and over what period. For instance, if fever was an inclusion criteria how come only a proportion of children report fever by RSV detection status in Figure 3

Response 5: Thank you for raising this important point. Eligible children were those <5 years old at hospital admission who developed a fever or at least one respiratory symptom within the preceding 14 days and were screened within 72 hours of admission. Children who did not reportedly develop a fever qualified for eligibility by reportedly developing at least one respiratory symptom. In the original text, we had not clarified which respiratory symptoms we considered. Therefore, in response to your feedback, we added the following text to the “*Study Design and Population*” subsection of the “*Methods*” section:

“Respiratory symptoms qualifying for eligibility included cough, earache, runny nose or nasal congestion, sore throat, posttussive vomiting, wheezing, rapid or shallow breathing, chest retractions or abdominal breathing, stridor, apnea (including brief resolved unexplained events), and myalgia.”

Also, the authors may want to consider omitting p values for the baseline variables in Table 1, which are superfluous and not necessarily reflecting meaningful clinical differences in some cases (e.g. exposure to smoking 84 vs 81%) within the sample size.

Response 6: We appreciate your thoughtful observation. We agree that some observed baseline differences (e.g., any tobacco-related smoke exposure of 84.3% vs. 80.8%) are unlikely to reflect clinically meaningful distinctions. At the same time, we believe that presenting a measure of the statistical strength of evidence can help the reader assess the degree of imbalance in a standardized manner. We therefore report *p*-values alongside measures of magnitude not to conflate statistical and clinical significance, but to equip readers with the information necessary to interpret these differences. To improve the robustness of our findings, we have systematically omitted from **Table 1** *p*-values for comparisons involving sparse data (i.e., any expected cell count <5).

In addition, to further aid interpretation, we have added analyses of in-hospital outcomes, adjusting for key potential confounders (**Table 3**). The adjusted estimates provide directly comparable measures of association and place greater emphasis on outcomes most relevant to clinical care, thereby complementing the descriptive comparisons in **Table 1**.

Re: Spectrum01727-25R1 (Epidemiology of respiratory syncytial virus in young, hospitalized children in Jordan: A prospective viral surveillance study)

Dear Dr. Justin Z. Amarin:

I am pleased to inform you that your manuscript has been accepted, and I am forwarding it to the ASM production staff for publication. Your paper will first be checked to make sure all elements meet the technical requirements. ASM staff will contact you if anything needs to be revised before copyediting and production can begin. Otherwise, you will be notified when your proofs are ready to be viewed.

Please note there are 3 typos/formatting errors that should be checked and corrected as required during the proof process (see reviewer comments below). Otherwise I would recommend that you order symptoms by cluster (e.g. lower respiratory, systemic) as opposed to alphabetically in Figure 3.

Sincerely,
Alex Dulovic
Editor
Microbiology Spectrum

Reviewer #1 (Comments for the Author):

This is a well-conducted, robust, and highly relevant prospective viral surveillance study. The authors present compelling data on the significant burden and clinical impact of Respiratory Syncytial Virus (RSV) among hospitalized young children in Jordan. The manuscript is clearly written, the methodology is sound, and the statistical analyses are appropriate. The study's major strength lies in its longitudinal design within a sentinel surveillance site, allowing for a powerful comparison of post-pandemic seasonality with historical patterns—a crucial contribution to the field, especially from the under-represented Eastern Mediterranean region. The findings have immediate and important implications for public health policy and the timing of preventive interventions in Jordan and similar settings.

The revisions have significantly strengthened the manuscript. The responses to previous feedback are thoughtful and the incorporation of additional data, particularly the adjusted analyses and the more nuanced discussion of co-detections and historical context, has greatly enhanced the paper's depth and impact.

I recommend acceptance of the manuscript after addressing the following minor points and suggestions for clarification.

Major Comments:

Viral Co-detections and Clinical Interpretation (Page 10-11, Lines 228-278):

The discussion on viral co-detection is much improved. However, given that over a third of RSV cases had a co-detection, I encourage a slightly deeper discussion on the potential clinical implications. While you rightly state that the significance "remains to be fully elucidated," you could briefly hypothesize. For instance, does co-detection with HRV (the most common) potentially lead to more severe disease through immune modulation, or could it simply be a frequent bystander? A sentence or two referencing the ongoing debate (e.g., additive severity vs. viral interference) would add valuable perspective for the reader. The citation of your group's work on selection bias (Ref 22) is an excellent addition here.

Antibiotic Use (Page 10, Lines 247-253):

The near-ubiquitous use of antibiotics (97.6% in RSV+ children) is a staggering and critically important finding. The link to diagnostic uncertainty is well-made. I suggest strengthening this point by explicitly stating that this pattern represents a major opportunity for antimicrobial stewardship. The high rate of antibiotic use, despite a confirmed viral etiology in a large majority of cases, underscores a significant gap in clinical practice that has direct consequences for antimicrobial resistance. This is a powerful public health message that deserves emphasis.

Minor Comments:

Abstract (Page 2, Line 33-35):

The abstract states RSV was the "second most common virus overall." While accurate, given the high frequency of HRV, it might be slightly misleading without the context that HRV PCR is cross-reactive with other enteroviruses (as correctly noted in the Methods, Page 7, Lines 138-140). Consider adding a brief qualifier in the abstract, e.g., "...second most common virus overall (after human rhinovirus/enterovirus)..."

Methods - Laboratory Testing (Page 7, Line 140):

The note on HRV/enterovirus cross-reactivity is perfect. For absolute clarity, you might consider specifying the molecular target of your assay (e.g., 5' UTR), as this is the region responsible for the cross-reactivity.

Results - Baseline Characteristics (Page 9, Line 200):

The very low rate of palivizumab use (0.2%) is noted. It would be useful to know if any of these 5 children were in the RSV-positive group. The table footnote (Table 1) states no inferential statistics were computed due to sparsity, but the raw numbers (5 in non-RSV vs. 0 in RSV) could still be presented in the table for transparency.

Discussion - Seasonality (Page 11, Lines 259-269):

The discussion on the return to winter seasonality is excellent. When mentioning the "multinational analysis" (Ref 21), it would be helpful to briefly note its key conclusion regarding the Eastern Mediterranean's under-representation, which makes your study's contribution even more vital. E.g., "...a recent multinational analysis demonstrated... albeit with little representation from the Eastern Mediterranean region, a gap which our study directly addresses."

References:

The reference list is comprehensive and appropriate. All citations appear to be accurate and relevant.

Typos and Formatting:

Page 19, Table 1: The row for "Daycare or preschool attendance" has a formatting error in the "Overall" column (932,608 (3.6)). This should likely be 92/2,608 (3.5%) or similar. Please check the numerator and denominator for this row and the "Any tobacco..." row for consistency.

Page 21, Table 2: The abbreviation for "Invasive mechanical ventilation" is listed as "IMV" in the table header but spelled out in the key. This is fine, but for consistency, ensure all abbreviations are defined in the key (CPAP/BIPAP, ICU, IMV are all defined correctly).

Page 26, Figure 3: The p-value for "Seizure" is listed as <0.001. The bar graph appears to show a very small, non-significant difference. Please verify the correctness of this p-value.

This is an outstanding piece of epidemiological research. The authors are to be commended for their rigorous work and for their thoughtful revisions. The manuscript provides essential data that will be valuable to researchers, clinicians, and public health officials globally, particularly in resource-limited settings. I have no further substantial objections and believe the manuscript is suitable for publication after these minor revisions.